# Faster Non-Convex Federated Learning via Global and Local Momentum

**Rudrajit Das**[1]    **Anish Acharya** [*1]    **Abolfazl Hashemi** [*2]    **Sujay Sanghavi**[1]    **Inderjit S. Dhillon**[1]    **Ufuk Topcu**[1]

[1]University of Texas at Austin, USA
[2]Purdue University, West Lafayette, Indiana, USA

## Abstract

We propose `FedGLOMO`, a novel federated learning (FL) algorithm with an iteration complexity of $\mathcal{O}(\epsilon^{-1.5})$ to converge to an $\epsilon$-stationary point (i.e., $\mathbb{E}[\|\nabla f(x)\|^2] \leq \epsilon$) for smooth non-convex functions – under arbitrary client heterogeneity and compressed communication – compared to the $\mathcal{O}(\epsilon^{-2})$ complexity of most prior works. Our key algorithmic idea that enables achieving this improved complexity is based on the observation that the convergence in FL is hampered by two sources of high variance: (i) the global server aggregation step with multiple local updates, exacerbated by client heterogeneity, and (ii) the noise of the local client-level stochastic gradients. The first issue is particularly detrimental to FL algorithms that perform plain averaging at the server. By modeling the server aggregation step as a generalized gradient-type update, we propose a variance-reducing momentum-based global update at the server, which when applied in conjunction with variance-reduced local updates at the clients, enables `FedGLOMO` to enjoy an improved convergence rate. Our experiments illustrate the intrinsic variance reduction effect of `FedGLOMO`, which implicitly suppresses client-drift in heterogeneous data distribution settings and promotes communication efficiency.

## 1 INTRODUCTION

Federated learning (FL) is a new edge-computing approach that advocates training statistical models directly on remote devices by leveraging enhanced local resources on each device (McMahan et al. [2017]). In a standard FL setting, there are $n$ clients, each having its own training data, and

---
[*]Equal Contribution

a central server that is trying to train a model, parameterized by $\boldsymbol{w} \in \mathbb{R}^d$, using the clients' data. Suppose the data distribution of the $i^{\text{th}}$ client is $\mathcal{D}_i$. Then the $i^{\text{th}}$ client has an objective function $f_i(\boldsymbol{w})$ which is the expected loss, with respect to some loss function $\ell$, over data drawn from $\mathcal{D}_i$, and the goal of the central server is to optimize the average [1] loss $f(\boldsymbol{w})$, over the $n$ clients, i.e.,

$$f(\boldsymbol{w}) := \frac{1}{n} \sum_{i=1}^{n} f_i(\boldsymbol{w}) \ \& \ f_i(\boldsymbol{w}) = \mathbb{E}_{\boldsymbol{x} \sim \mathcal{D}_i}[\ell(\boldsymbol{x}, \boldsymbol{w})]. \quad (1)$$

The setting where the data distributions of all the clients are identical, i.e. $\mathcal{D}_1 = \ldots = \mathcal{D}_n$, is typically known as the "homogeneous" setting. Otherwise, the settings where the data distributions are *not* identical are referred to as the "heterogeneous" settings.

The core algorithmic idea of FL – in the form of `FedAvg` – was introduced in McMahan et al. [2017]. In `FedAvg` (summarized in Algorithm 3), a *subset* of the clients perform *multiple* steps of gradient descent based updates on their local data and then communicate back their respective updates to the server, which then averages them to update the global model (hence the name `FedAvg`). This idea of performing multiple local updates before averaging once reduces the communication cost required for training. Another essential strategy in FL to cut down the communication cost is to have the clients send compressed/quantized messages to the server in every round – this is of particular significance for training deep learning models where the number of model parameters is in millions or more.

In practice however, performing multiple local updates on clients with *heterogeneous* data distributions leads to the so-called phenomenon of "client drift", wherein the individual client updates do not align well (due to over-fitting on the local client data) inhibiting the convergence of `FedAvg` to the optimum of the average loss over all the clients. In this paper, we identify the high variance associated with the

---
[1]In general this may be a weighted average, but here we only consider uniform weights, i.e., each weight is $1/n$.

*Accepted for the 38th Conference on Uncertainty in Artificial Intelligence* (UAI 2022).

simple averaging step of `FedAvg` for the global update to be at the heart of this issue.

Ever since the development of FL, significant attention has been devoted to analyzing `FedAvg` under different settings, modifying `FedAvg` using ideas from centralized optimization to accelerate the training or to reduce the communication cost; we discuss these works in Section 2. Compared to centralized optimization, a formidable challenge in the theoretical analysis of FL algorithms is the use of multiple local updates in the clients which is compounded by the *heterogeneous* nature of data distribution among the clients. To limit the extent of client heterogeneity, a standard assumption in FL theory is the *bounded client dissimilarity (BCD) assumption*, i.e.,

$$\frac{1}{n} \sum_{i=1}^{n} \|\nabla f_i(\boldsymbol{w}) - \nabla f(\boldsymbol{w})\|^2 \leq G^2 \ \forall \ \boldsymbol{w}, \qquad (2)$$

for some large enough constant $G < \infty$ (e.g., see A1 in Karimireddy et al. [2020]). But this assumption is limiting as it does not allow for *arbitrarily large client heterogeneity*.

Recently, Arjevani et al. [2019] showed that the stochastic first-order complexity of any algorithm in the *centralized setting* to reach an $\epsilon$-stationary point (i.e., $\mathbb{E}[\|\nabla f(\boldsymbol{x})\|^2] \leq \epsilon$) for *smooth non-convex functions* is $\Omega(\epsilon^{-1.5})$. It is well known that vanilla SGD has a suboptimal complexity of $\mathcal{O}(\epsilon^{-2})$ as it cannot mitigate the high variance of the stochastic gradient noise. Recognizing this issue, *variance-reducing* techniques for SGD (Fang et al. [2018], Zhou et al. [2018], Cutkosky and Orabona [2019], Liu et al. [2020]) have been proposed that attain the optimal complexity of $\mathcal{O}(\epsilon^{-1.5})$. Coming to the federated setting, as we discuss in this paper, in addition to the noise in the *local* client-level stochastic gradients, one has to also contend with the high variance associated with the *global* server aggregation step which depends on the client heterogeneity and the number of local update steps. In this case, as we argue in the subsequent sections, applying only local client-level variance-reduction is not enough for improving the iteration complexity of vanilla `FedAvg` beyond $\mathcal{O}(\epsilon^{-2})$ for smooth, non-convex losses.

To alleviate the issue of variance due to heterogeneity, we propose a novel FL algorithm with *compressed communication* called `FedGLOMO` (Algorithm 1 and 2) which applies G*lobal* as well as LO*cal variance-reducing* MO*mentum* to the server update and client updates, respectively. We prove that the iteration complexity of `FedGLOMO` is $\mathcal{O}(\epsilon^{-1.5})$ in the smooth non-convex case, which is better than the $\mathcal{O}(\epsilon^{-2})$ complexity of related works in the FL setting; see Table 1 and Theorem 1. Further, our theory does not use the BCD assumption, i.e. eq. (2), which is a standard assumption in related works. Instead, we propose and use Assumption 4, which is a more realistic and *empirically verified* assumption on the client drift, even allowing for arbitrary client heterogeneity. It is worth mentioning here that for FL, Karimireddy et al. [2020] also propose an algorithm (`MimeMVR`) which

is shown to attain this improved complexity of $\mathcal{O}(\epsilon^{-1.5})$ but *with* the BCD assumption and *no* compressed communication; we talk about this at the end of Section 2.

We summarize our **contributions** next:

**(a)** We propose `FedGLOMO` (Alg. 1 and 2), in which we apply a *novel global momentum term at the server* in addition to *local momentum at the clients*. The design of `FedGLOMO` is motivated by two critical issues that need to be alleviated to accelerate convergence in FL; these are the high variances associated with: (i) the *global* server aggregation step due to heterogeneity of clients when there are multiple local updates, and (ii) the noise of *local* client-level stochastic gradients. Global and local momentum result in *variance reduction* for the global server update and the local client updates, allowing us to tackle (i) and (ii), respectively. This enables `FedGLOMO` to converge to an $\epsilon$-stationary point (i.e., $\mathbb{E}[\|\nabla f(\boldsymbol{x})\|^2] \leq \epsilon$) for smooth non-convex functions in $\mathcal{O}(\epsilon^{-1.5})$ gradient-based updates, which is better than the $\mathcal{O}(\epsilon^{-2})$ complexity of most related works in the FL setting; see Table 1 and Theorem 1.

**(b)** Unlike prior work, our theory does not use the limiting bounded client dissimilarity assumption (i.e., eq. (2)). Instead, to tighten our result, we propose and use Assumption 4 – which is a novel assumption on the client drift, even allowing for *arbitrary client heterogeneity* in the worst case. We empirically verify that Assumption 4 holds for `FedGLOMO` as well as `FedAvg`. Theoretically, we also show that Assumption 4 holds for *any* FL algorithm in the case of linear regression and also with networks whose training dynamics follow that of a linearized model (a.k.a. the "NTK" regime). Refer to the discussion after Assumption 4 and Remark 2 for details.

**(c)** `FedGLOMO` is the *first FL algorithm* achieving $\mathcal{O}(\epsilon^{-1.5})$ complexity while allowing *compressed client-to-server communication*. We emphasize that from the theory perspective, applying compression in `FedGLOMO` is not trivial and the most obvious approach does not work; see Remark 3.

**(d)** In Section 6, experiments on CIFAR-10 and Fashion-MNIST (Xiao et al. [2017]) show that in a highly heterogeneous setting of at most two (out of ten) classes per client, `FedGLOMO` requires only about *one-third* the number of bits used by `FedAvg` with PyTorch's default momentum applied to the local client updates; see Figure 1. Our experiments also illustrate the variance reduction provided by our scheme which implicitly mitigates client-drift under heterogeneous data distribution and in turn promotes communication-efficiency.

## 2 RELATED WORK

**`FedAvg` and related methods:** Reisizadeh et al. [2020] propose `FedPAQ` which is basically `FedAvg` (McMahan

et al. [2017]) with quantized client-to-server communication, and establish its convergence for the homogeneous case. Li et al. [2019] establish the convergence of `FedAvg` for strongly convex functions with heterogeneity (assuming bounded client dissimilarity) but without any compressed communication. Haddadpour et al. [2021] propose `FedCOMGATE` which incorporates gradient tracking (Pu and Nedić [2020]) and derive results with data heterogeneity and quantized communication. Karimireddy et al. [2019] propose `SCAFFOLD` which uses control-variates to mitigate the client-drift owing to the heterogeneity of clients. Li et al. [2018] present `FedProx` which adds a proximal term to control the deviation of the client parameters from the global server parameter in the previous round. Reddi et al. [2020] propose federated versions of commonly used adaptive optimization methods and prove their convergence under heterogeneity. Local SGD (Zinkevich et al. [2010], Stich [2018], Yu et al. [2018], Wang and Joshi [2018], Basu et al. [2019], Stich and Karimireddy [2019], Patel and Dieuleveut [2019], Woodworth et al. [2020], Bayoumi et al. [2020], Liang et al. [2019], Koloskova et al. [2020]) is very similar to FL and is essentially based on the same principle as `FedAvg`. However, in local SGD, there is usually no data heterogeneity and all the clients participate in each round (known as "full device participation"), both of which do not hold in FL and simplify the derivation of convergence results.

Wang et al. [2019], Huo et al. [2020] present momentum-based updates at the server without any improvement in the convergence rate as compared to momentum-free updates. Qu et al. [2020] present Nesterov accelerated `FedAvg` for convex objectives. Karimireddy et al. [2020] propose `Mime(MVR)` which applies momentum at the client-level based on globally computed statistics to control client-drift. Khanduri et al. [2021] propose `STEM` which applies momentum globally and locally for local SGD; however, their server aggregation step is just plain averaging as they do not have deal with server-side variance reduction, since all the clients participate in local SGD.

**Distributed optimization with compression:** References Alistarh et al. [2017], Suresh et al. [2017], Reisizadeh et al. [2020], Haddadpour et al. [2021], Tang et al. [2018], Wu et al. [2018], Bernstein et al. [2018], Alistarh et al. [2018], Lin et al. [2017], Stich et al. [2018], Basu et al. [2019], Hashemi et al. [2021], Chen et al. [2020, 2021] aim to minimize the communication bottleneck in distributed optimization by transmitting compressed messages to the central server and establishing their convergence. Horváth et al. [2019], Gorbunov et al. [2021] provide distributed algorithms with improved convergence rates by also applying variance reduction and periodically using full gradients; however, there are no multiple local updates in these works. In Appendix D, we compare our work's complexity against that of Gorbunov et al. [2021]. In this work, we employ the quantization operator of Alistarh et al. [2017].

**Complexity for smooth non-convex stochastic optimization:** Arjevani et al. [2019] show that the optimal stochastic first-order complexity to reach an $\epsilon$-stationary point (i.e., $\mathbb{E}[\|\nabla f(\boldsymbol{x})\|^2] \le \epsilon$) is $\mathcal{O}(\frac{\sigma}{\epsilon^{1.5}})$ where $\sigma^2$ is the variance of the stochastic gradients. Unfortunately, vanilla SGD is sub-optimal and *variance-reducing* techniques must be applied to attain the optimal complexity; some noteworthy works on variance-reduction for SGD are `SVRG` (Johnson and Zhang [2013]), `SAGA` (Defazio et al. [2014]) and `SARAH` (Nguyen et al. [2017]). SVRG-style algorithms such as `SPIDER` (Fang et al. [2018]) and `SNVRG` (Zhou et al. [2018]) attain this optimal complexity by periodically using giant batch sizes. Cutkosky and Orabona [2019] propose `STORM` which also attains this optimal complexity with adaptive learning rates, but without using any large batches. The key idea of `STORM` is momentum-based variance reduction, obtained by using the stochastic gradient at the previous point *computed over the same batch* on which the stochastic gradient at the current point is computed. Liu et al. [2020] present a much simpler proof for essentially the same algorithm by employing a constant learning rate and requiring a large batch size only at the first iteration. Our key idea of global and local momentum is `STORM`-like *variance-reducing* momentum applied to the aggregation step at the server, interpreted as a generalized gradient-type update, and the local client updates, respectively; see Section 4.

Table 1 compares the complexities of the most relevant related works in FL ($r < n$) and local SGD ($r = n$) with ours on smooth non-convex functions. Note that under the more challenging FL setting with partial-device participation, only `FedGLOMO` and `MimeMVR` (Karimireddy et al. [2020]) attain the improved iteration complexity of $\mathcal{O}(\epsilon^{-1.5})$ with respect to $\epsilon$. However, unlike Karimireddy et al. [2020], our work does not rely on the bounded client dissimilarity assumption (eq. (2)) and allows for compressed client-to-server communication, in which case maintaining the improved complexity is not trivial; for details, see Remarks 2 and 3, respectively. There are meaningful algorithmic differences between our work and Karimireddy et al. [2020] too. The biggest one is that while we explicitly apply momentum in the server aggregation step (global momentum) as well as in the client updates (local momentum), Karimireddy et al. [2020] only apply *globally computed* momentum in the local client updates. For a detailed discussion of the differences of our work from Karimireddy et al. [2020], see Appendix C. Since `Mime` is designed to deal with client drift, we empirically compare it against `FedGLOMO` without compression in a highly heterogeneous setting in Section 6.

## 3 PRELIMINARIES

Recall the setting and the optimization problem that the server is trying to solve as defined in eq. (1). We assume that the clients have access to unbiased stochastic gradients

Table 1: Number of gradient updates, i.e., $T$, required to achieve $\mathbb{E}[\|\nabla f(\boldsymbol{w})\|^2] \leq \epsilon$ on smooth non-convex functions. Here, $n$ is the total number of clients and $r$ is the number of clients participating in each round. "Client Participation" asks whether all $(r = n)$ or only a subset $(r < n)$ of the clients participate in each round. "BCD?" asks if the bounded client dissimilarity assumption (eq. (2)) is used or not. "Compression?" asks whether compressed communication is involved or not. $*1$: $\alpha \leq n$ is a problem-dependent quantity; in practice, we expect $\alpha \ll n$ as confirmed in our experiments.

| Ref. | $T$ | Client Participation | BCD? | Compression? |
|---|---|---|---|---|
| Koloskova et al. [2020], Wang et al. [2019] | $\mathcal{O}(\frac{1}{n\epsilon^2})$ | Full $(r = n)$ | Yes | ✗ |
| Haddadpour et al. [2021] | $\mathcal{O}(\frac{1}{n\epsilon^2})$ | Full $(r = n)$ | Yes | ✓ |
| Khanduri et al. [2021] | $\mathcal{O}(\frac{1}{n\epsilon^{1.5}})$ | Full $(r = n)$ | Yes | ✗ |
| Karimireddy et al. [2019] | $\mathcal{O}(\frac{1}{r\epsilon^2})$ | Partial $(r < n)$ | Yes | ✗ |
| Karimireddy et al. [2020] | $\mathcal{O}(\frac{1}{\sqrt{r}\epsilon^{1.5}})$ | Partial $(r < n)$ | Yes | ✗ |
| **This work** (FedGLOMO) | $\mathcal{O}\big(\max\big(\sqrt{\frac{\alpha}{n}}, \frac{1}{\sqrt{r}}\big)\frac{1}{\epsilon^{1.5}}\big)^{*1}$ | Partial $(r < n)$ | **No** | ✓ |

of their individual losses. We denote the stochastic gradient of $f_i$ at $\boldsymbol{w}$ computed over a batch of samples $\mathcal{B}$, by $\widetilde{\nabla} f_i(\boldsymbol{w}; \mathcal{B})$. Also in this paper, $K$ is the number of communication rounds, $E$ is the number of local updates per round or the period, and $T = KE$ is the total number of local updates or the (order-wise) number of gradient-based updates. Further, $r$ is the number of clients that the server accesses in each round, i.e., the global batch size.

Vectors and matrices are written in boldface. For any positive integer $m$, the set $\{1, \ldots, m\}$ is denoted by $[m]$, and the uniform distribution over the set $\{0, \ldots, m\}$ is denoted by $\text{Unif}[0, m]$. $\mathbb{1}(.)$ is the indicator function. Next, we recap smooth functions.

**Definition 3.1 (Smoothness).** A function $g : \Theta \to \mathbb{R}$ is to said to be $L$-smooth if for all $\boldsymbol{\theta}, \boldsymbol{\theta}' \in \Theta$, $\|\nabla g(\boldsymbol{\theta}) - \nabla g(\boldsymbol{\theta}')\| \leq L\|\boldsymbol{\theta} - \boldsymbol{\theta}'\|$. For all $\boldsymbol{\theta}, \boldsymbol{\theta}' \in \Theta$, we also have: $g(\boldsymbol{\theta}') \leq g(\boldsymbol{\theta}) + \langle \nabla g(\boldsymbol{\theta}), \boldsymbol{\theta}' - \boldsymbol{\theta}\rangle + \frac{L}{2}\|\boldsymbol{\theta}' - \boldsymbol{\theta}\|^2$.

## 4 FEDGLOMO: GLOBAL AND LOCAL MOMENTUM-BASED VARIANCE REDUCTION

There are two issues that need to be alleviated for improving the convergence rate in FL: (i) the high variance of simple averaging used in the *global* server aggregation step (of FedAvg), when there are multiple local updates, which is exacerbated by heterogeneity of the clients, and (ii) the high variance associated with the noise of *local* client-level stochastic gradients. The key idea of FedGLOMO (Algorithm 1 and 2) is to apply *variance-reducing* **global** and **local** momentum to combat (i) and (ii), respectively. We now describe global and local momentum in detail.

**Global** momentum is applied to the sever aggregation step which is line 10 in Algorithm 1. To understand it better, let us revisit FedAvg (summarized in Algorithm 3, although in a

---

**Algorithm 1** FedGLOMO - Server Update

1: **Input:** Initial point $\boldsymbol{w}_0$, # of rounds of communication $K$, period $E$, learning rates $\{\eta_k\}_{k=0}^{K-1}$ and global batch size $r$. $Q_D$ is the quantization operator. Set $\boldsymbol{w}_{-1} = \boldsymbol{w}_0$.

2: **for** $k = 0, \ldots, K-1$ **do**
3:     Server sends $\boldsymbol{w}_k, \boldsymbol{w}_{k-1}$ to a set $\mathcal{S}_k$ of $r$ clients chosen uniformly at random w/o replacement.
4:     **for** client $i \in \mathcal{S}_k$ **do**
5:         Set $\boldsymbol{w}_{k,0}^{(i)} = \boldsymbol{w}_k$ and $\widehat{\boldsymbol{w}}_{k-1,0}^{(i)} = \boldsymbol{w}_{k-1}$. Run Algorithm 2 for client $i$.
6:     **end for**
7:     **if** $k = 0$ **then**
8:         Set $\boldsymbol{u}_k = \frac{1}{r}\sum_{i \in \mathcal{S}_k} Q_D(\boldsymbol{w}_k - \boldsymbol{w}_{k,E}^{(i)})$.
9:     **else**
10:         Set $\boldsymbol{u}_k = \frac{\beta_k}{r}\sum_{i \in \mathcal{S}_k} Q_D(\boldsymbol{w}_k - \boldsymbol{w}_{k,E}^{(i)}) + (1 - \beta_k)\boldsymbol{u}_{k-1} + \frac{(1-\beta_k)}{r}\sum_{i \in \mathcal{S}_k} Q_D((\boldsymbol{w}_k - \boldsymbol{w}_{k,E}^{(i)}) - (\boldsymbol{w}_{k-1} - \widehat{\boldsymbol{w}}_{k-1,E}^{(i)}))$. // (Global Momentum)
11:     **end if**
12:     Update $\boldsymbol{w}_{k+1} = \boldsymbol{w}_k - \boldsymbol{u}_k$.
13: **end for**

---

slightly different way than usual) and its server aggregation step (line 12) which is just simple averaging. Similar to the update of SGD suffering from high variance, this naive averaging step – which we think of as the average of a batch of generalized stochastic gradients – is characterized by high variance stemming from heterogeneity and multiple local updates. So, this way of server aggregation slows down the convergence rate of FedAvg (and other related methods).

In this paper, we re-envision the server aggregation as a generalized gradient-based update by thinking of $(\boldsymbol{w}_k - \boldsymbol{w}_{k,E}^{(i)})$ as the generalized gradient. Then, we wish to incorporate the style of variance-reducing momentum applied in STORM (Cutkosky and Orabona [2019], Liu et al. [2020]) to our gen-

**Algorithm 2** FedGLOMO - Client Update

1: **for** $\tau = 0, \ldots, E-1$ **do**
2:   **if** $\tau = 0$ **then**
3:     Set $\boldsymbol{v}_{k,\tau}^{(i)} = \nabla f_i(\boldsymbol{w}_{k,\tau}^{(i)})$, $\widehat{\boldsymbol{v}}_{k-1,\tau}^{(i)} = \nabla f_i(\widehat{\boldsymbol{w}}_{k-1,\tau}^{(i)})$.
4:   **else**
5:     Pick a random batch of samples in client $i$, say $\mathcal{B}_{k,\tau}^{(i)}$. Compute the stochastic gradients of $f_i$ at $\boldsymbol{w}_{k,\tau}^{(i)}$, $\widehat{\boldsymbol{w}}_{k-1,\tau}^{(i)}$, $\boldsymbol{w}_{k,\tau-1}^{(i)}$ and $\widehat{\boldsymbol{w}}_{k-1,\tau-1}^{(i)}$ over $\mathcal{B}_{k,\tau}^{(i)}$ viz. $\widetilde{\nabla} f_i(\boldsymbol{w}_{k,\tau}^{(i)}; \mathcal{B}_{k,\tau}^{(i)})$, $\widetilde{\nabla} f_i(\widehat{\boldsymbol{w}}_{k-1,\tau}^{(i)}; \mathcal{B}_{k,\tau}^{(i)})$, $\widetilde{\nabla} f_i(\boldsymbol{w}_{k,\tau-1}^{(i)}; \mathcal{B}_{k,\tau}^{(i)})$ and $\widetilde{\nabla} f_i(\widehat{\boldsymbol{w}}_{k-1,\tau-1}^{(i)}; \mathcal{B}_{k,\tau}^{(i)})$.
6:     Update: $\boldsymbol{v}_{k,\tau}^{(i)} = \widetilde{\nabla} f_i(\boldsymbol{w}_{k,\tau}^{(i)}; \mathcal{B}_{k,\tau}^{(i)}) + (\boldsymbol{v}_{k,\tau-1}^{(i)} - \widetilde{\nabla} f_i(\boldsymbol{w}_{k,\tau-1}^{(i)}; \mathcal{B}_{k,\tau}^{(i)}))$ and $\widehat{\boldsymbol{v}}_{k-1,\tau}^{(i)} = \widetilde{\nabla} f_i(\widehat{\boldsymbol{w}}_{k-1,\tau}^{(i)}; \mathcal{B}_{k,\tau}^{(i)}) + (\widehat{\boldsymbol{v}}_{k-1,\tau-1}^{(i)} - \widetilde{\nabla} f_i(\widehat{\boldsymbol{w}}_{k-1,\tau-1}^{(i)}; \mathcal{B}_{k,\tau}^{(i)}))$. // (Local Mom.)
7:   **end if**
8:   Update $\boldsymbol{w}_{k,\tau+1}^{(i)} = \boldsymbol{w}_{k,\tau}^{(i)} - \eta_k \boldsymbol{v}_{k,\tau}^{(i)}$ and $\widehat{\boldsymbol{w}}_{k-1,\tau+1}^{(i)} = \widehat{\boldsymbol{w}}_{k-1,\tau}^{(i)} - \eta_k \widehat{\boldsymbol{v}}_{k-1,\tau}^{(i)}$.
9: **end for**
10: Send $Q_D(\boldsymbol{w}_k - \boldsymbol{w}_{k,E}^{(i)})$ and $Q_D((\boldsymbol{w}_k - \boldsymbol{w}_{k,E}^{(i)}) - (\boldsymbol{w}_{k-1} - \widehat{\boldsymbol{w}}_{k-1,E}^{(i)}))$ to the server.

---

**Algorithm 3** FedAvg McMahan et al. [2017]

1: **Input:** Initial point $\boldsymbol{w}_0$, # of communication rounds $K$, period $E$, learning rates $\{\eta_k\}_{k=0}^{K-1}$ and global batch size $r$.
2: **for** $k = 0, \ldots, K-1$ **do**
3:   Server sends $\boldsymbol{w}_k$ to a set $\mathcal{S}_k$ of $r$ clients chosen uniformly at random w/o replacement.
4:   **for** client $i \in \mathcal{S}_k$ **do**
5:     Set $\boldsymbol{w}_{k,0}^{(i)} = \boldsymbol{w}_k$.
6:     **for** $\tau = 0, \ldots, E-1$ **do**
7:       Pick a random batch of samples in client $i$, $\mathcal{B}_{k,\tau}^{(i)}$. Compute the stochastic gradient of $f_i$ at $\boldsymbol{w}_{k,\tau}^{(i)}$ over $\mathcal{B}_{k,\tau}^{(i)}$, viz. $\widetilde{\nabla} f_i(\boldsymbol{w}_{k,\tau}^{(i)}; \mathcal{B}_{k,\tau}^{(i)})$.
8:       Update $\boldsymbol{w}_{k,\tau+1}^{(i)} = \boldsymbol{w}_{k,\tau}^{(i)} - \eta_k \widetilde{\nabla} f_i(\boldsymbol{w}_{k,\tau}^{(i)}; \mathcal{B}_{k,\tau}^{(i)})$.
9:     **end for**
10:     Send $(\boldsymbol{w}_k - \boldsymbol{w}_{k,E}^{(i)})$ to the server.
11:   **end for**
12:   Update $\boldsymbol{w}_{k+1} = \boldsymbol{w}_k - \frac{1}{r} \sum_{i \in \mathcal{S}_k} (\boldsymbol{w}_k - \boldsymbol{w}_{k,E}^{(i)})$.
13: **end for**

---

eralized gradient-based update; note that their method is for stochastic gradients in the case of centralized optimization. To that end, let us briefly recap STORM's update rule. For a function $h(\boldsymbol{z})$, STORM's update for the $j^{\text{th}}$ iteration is:

$$\boldsymbol{z}_{j+1} = \boldsymbol{z}_j - \eta_j \boldsymbol{v}_j, \text{ where } \boldsymbol{v}_j = \{\widetilde{\nabla} h(\boldsymbol{z}_j; \xi_j) + (1-\beta_j)(\boldsymbol{v}_{j-1} - \widetilde{\nabla} h(\boldsymbol{z}_{j-1}; \xi_j))\mathbb{1}(j > 0)\}. \quad (3)$$

In eq. (3), $\xi_j$ denotes the source of randomness in the $j^{\text{th}}$ iteration and $\beta_j \in [0,1)$ is the momentum parameter. Note the use of the stochastic gradient at $\boldsymbol{z}_{j-1}$ computed on $\xi_j$. Coming back to Algorithm 1, the quantity $\boldsymbol{u}_k$ plays the role of $\boldsymbol{v}_j$ in eq. (3). To see this clearly, let us analyze $\mathbb{E}_{Q_D}[\boldsymbol{u}_k]$ (see lines 8 and 10 in Algorithm 1). Under Assumption 3, the compression operator $Q_D$ produces an unbiased estimate of the input. Then defining $g(\boldsymbol{w}_k; \mathcal{S}_k) \triangleq \frac{1}{r} \sum_{i \in \mathcal{S}_k} (\boldsymbol{w}_k - \boldsymbol{w}_{k,E}^{(i)})$ and $\widehat{g}(\boldsymbol{w}_{k-1}; \mathcal{S}_k) \triangleq \frac{1}{r} \sum_{i \in \mathcal{S}_k} (\boldsymbol{w}_{k-1} - \widehat{\boldsymbol{w}}_{k-1,E}^{(i)})$, we have:

$$\mathbb{E}_{Q_D}[\boldsymbol{u}_k] = \{g(\boldsymbol{w}_k; \mathcal{S}_k) + (1-\beta_k)(\boldsymbol{u}_{k-1} - \widehat{g}(\boldsymbol{w}_{k-1}; \mathcal{S}_k))\mathbb{1}(k > 0)\}. \quad (4)$$

In eq. (4), $g(\boldsymbol{w}_k; \mathcal{S}_k)$ and $\widehat{g}(\boldsymbol{w}_{k-1}; \mathcal{S}_k)$ play the roles of $\widetilde{\nabla} h(\boldsymbol{z}_j; \xi_j)$ and $\widetilde{\nabla} h(\boldsymbol{z}_{j-1}; \xi_j)$, respectively. With this, one can clearly see that eq. (4) is the analogue of eq. (3) for the global server aggregation in FL. However, this equivalence is not so apparent without looking at the expected value of $\boldsymbol{u}_k$ with respect to $Q_D$; in fact, the choice of quantities that are compressed in line 10 of Alg. 2 and used in line 10 of Alg. 1 is crucial for establishing provable guarantees (also see Remark 3).

Now that we understand global momentum, let us move on to **local** momentum. For this see lines 3, 6 and 8 in Algorithm 2; these give us $(\boldsymbol{w}_k - \boldsymbol{w}_{k,E}^{(i)})$ and $(\boldsymbol{w}_{k-1} - \widehat{\boldsymbol{w}}_{k-1,E}^{(i)})$ after running for $E$ steps. But notice that these lines are the same as eq. (3) with $\beta_j = 0$ and the stochastic gradient at the first iteration replaced by the full gradient. It is worth mentioning here that these local updates are also similar to SPIDER which is an SVRG-style update proposed in Fang et al. [2018]. However, recognizing that this is also a special case of the STORM update with $\beta_j = 0$, we prefer calling it momentum in order to have a unifying terminology for both the global and local updates.

One might wonder what is the role of global momentum as SPIDER can be extended to improve the complexity in distributed optimization *without multiple local updates*. For this, Appendix F, we consider FedLOMO (Algorithm 4 and 5 in the Appendix) which is a simpler version of FedGLOMO with only local momentum and *no* global momentum (i.e, plain averaging at the server which is equivalent to setting $\beta_k = 1$ in Algorithm 1), and show that it does not achieve $\mathcal{O}(\epsilon^{-1.5})$ complexity under partial-device participation and compression (see Theorem 3 in Appendix). The root cause of this is client heterogeneity which amplifies its effect under *multiple local updates*; without incorporating some form of variance reduction in the server aggregation step, the complexity cannot be improved.

Let us try to provide some intuition as to how incorporating global momentum helps. Suppose we keep $\eta_k = \eta$ and $\beta_k = \beta < 1$ for all $k$. Theoretically, we get a lower

bound for $\beta$ which is $\mathcal{O}(\eta^2)$. Then with this momentum-based aggregation strategy, the variance reduces by a factor of $\mathcal{O}(\beta/\eta) = \mathcal{O}(\eta)$ as compared to aggregation by plain averaging. (There are some other terms too but these are sufficiently small.) This reduction in the variance by a factor of $\mathcal{O}(\eta)$ is what improves the convergence rate of FedGLOMO.

It is true that FedGLOMO has to communicate twice the amount of information per round as compared to FedAvg or FedPAQ (Reisizadeh et al. [2020]) which is just FedAvg with compressed communication. One can set the precision of the quantizer sufficiently low to account for the extra per-round communication cost of FedGLOMO – we adopt this approach in our experiments. Also, we only assume access to the full client gradient in line 3 of Alg. 2 for simplicity of analysis, but our main result (i.e., Theorem 1) can be readily extended to the case of large enough batch sizes.

# 5 MAIN RESULT FOR FEDGLOMO

First, we state our assumptions.

**Assumption 1 (Smoothness).** $\ell(\boldsymbol{x}, \boldsymbol{w})$ *is $L$-smooth with respect to $\boldsymbol{w}$, for all $\boldsymbol{x}$. Thus, each $f_i(\boldsymbol{w})$ ($i \in [n]$) is $L$-smooth, and so is $f(\boldsymbol{w})$.*

**Assumption 2 (Non-negativity).** *Each $f_i(\boldsymbol{w})$ is non-negative and therefore, $f_i^* \triangleq \min f_i(\boldsymbol{w}) \geq 0$.*

Most loss functions used in practice satisfy this anyways and if not, we can just add a constant offset to achieve non-negativity.

**Assumption 3 (Quantization).** *The quantization operator $Q_D$ in Alg. 1 and 2 is unbiased, i.e., $\mathbb{E}[Q_D(\boldsymbol{x})|\boldsymbol{x}] = \boldsymbol{x}$, and its variance satisfies $\mathbb{E}[\|Q_D(\boldsymbol{x}) - \boldsymbol{x}\|^2|\boldsymbol{x}] \leq q\|\boldsymbol{x}\|^2$ for some $q > 0$. The "qsgd" operator proposed in Section 3.1 of Alistarh et al. [2017] satisfies Assumption 3.*

**Assumption 4 (Client Drift/Heterogeneity).** *Let $\mathcal{A}$ be an FL algorithm with $E$ local update steps and $K$ communication rounds. Let $\boldsymbol{w}_{k,\tau}^{(i)}$ be the $i^{th}$ client's local parameter at the start of the $(\tau+1)^{st}$ local step of the $(k+1)^{st}$ round of $\mathcal{A}$, for $i \in [n]$ (similar to the notation in Alg. 1, 2, and 3). Define $\widetilde{\boldsymbol{e}}_{k,\tau}^{(i)} \triangleq \nabla f_i(\boldsymbol{w}_{k,\tau}^{(i)}) - \nabla f_i\big(\frac{1}{n}\sum_{j \in [n]} \boldsymbol{w}_{k,\tau}^{(j)}\big)$. Then for some $\alpha \ll n$, the following holds:*

$$\mathbb{E}\Big[\Big\|\sum_{i \in [n]} \widetilde{\boldsymbol{e}}_{k,\tau}^{(i)}\Big\|^2\Big] \leq \alpha \sum_{i \in [n]} \mathbb{E}\Big[\Big\|\widetilde{\boldsymbol{e}}_{k,\tau}^{(i)}\Big\|^2\Big], \qquad (5)$$

$\forall \, \tau \in \{0, \ldots, E-1\}$ *and $k \in \{0, \ldots, K-1\}$. The expectation above is w.r.t. any stochasticity in the local updates.*

Equation (5) in the above assumption always holds with $\alpha = n$ for any FL algorithm; this follows from the fact that for any $m > 1$ vectors $\{\boldsymbol{a}_j\}_{j=1}^m$, $\|\sum_{j=1}^m \boldsymbol{a}_j\|^2 \leq$ $m\sum_{j=1}^m \|\boldsymbol{a}_j\|^2$ (this can be obtained by using the Cauchy-Schwarz inequality). However, we empirically observe $\alpha \ll n$ in practice for FedGLOMO as well as FedAvg; see Section 6 and Appendix H, respectively. The value of $\alpha$ in Assumption 4 is a measure of the amount of client drift induced by the algorithm which also depends on the degree of heterogeneity in the system – as the heterogeneity increases (decreases), we observe $\alpha$ to also increase (decrease).

From Figure 3 (in Section 6), we see that for the highly heterogeneous setting that we consider for our experiments in Section 6, $\alpha < 0.06n$ for most of the trajectory of FedGLOMO on both CIFAR-10 and Fashion-MNIST (abbreviated as FMNIST). In the homogeneous case, $\alpha < 0.03n$ and $\alpha < 0.02n$ for most of the trajectory on CIFAR-10 and FMNIST, respectively. We observe a similar trend of $\alpha$ for FedAvg in Appendix H. Additionally, we derive a convergence result for FedAvg under Assumption 4 and without the bounded client dissimilarity assumption (i.e., eq. (2)) in Appendix H.

**Some theoretical motivation for Assumption 4:** Let us consider *linear regression* to provide a scenario where $\alpha = 0$ provably for **any** FL algorithm. Suppose in client $i$, we have feature and label pairs $(\boldsymbol{x}, y) \sim (\mathcal{X}_i, \mathcal{Y}_i)$, where the label

$$y = \langle \boldsymbol{w}_i^*, \boldsymbol{x} \rangle + \xi,$$

with $\xi \sim \mathcal{N}_i$ being independent zero-mean client-dependent random noise. Obviously, the label distribution $\mathcal{Y}_i$ here depends on the feature distribution $\mathcal{X}_i$, noise distribution $\mathcal{N}_i$ and $\boldsymbol{w}_i^*$. We assume that the covariance matrix of the feature vectors is the same across all the clients, i.e., $\mathbb{E}_{\boldsymbol{x} \sim \mathcal{X}_i}[\boldsymbol{x}\boldsymbol{x}^T] = \boldsymbol{Q}$ for all $i \in [n]$; this is possible for e.g., by normalization or whitening of the features. Note that by assuming the same covariance matrix across all the clients, we are *not* assuming that the feature distributions are the same across clients, but even if they are, there is heterogeneity through the different label distributions. Then, with the squared loss, our per-client objective function is:

$$f_i(\boldsymbol{w}) = \mathbb{E}_{(\boldsymbol{x},y) \sim (\mathcal{X}_i, \mathcal{Y}_i)}\Big[\frac{1}{2}(y - \langle \boldsymbol{w}, \boldsymbol{x} \rangle)^2\Big].$$

With the aforementioned conditions, it can be verified that $\nabla f_i(\boldsymbol{w}) = \boldsymbol{Q}(\boldsymbol{w} - \boldsymbol{w}_i^*)$. Thus,

$$\widetilde{\boldsymbol{e}}_{k,\tau}^{(i)} = \boldsymbol{Q}\Big(\boldsymbol{w}_{k,\tau}^{(i)} - \frac{1}{n}\sum_{j \in [n]} \boldsymbol{w}_{k,\tau}^{(j)}\Big),$$

and so $\sum_{i \in [n]} \widetilde{\boldsymbol{e}}_{k,\tau}^{(i)} = \vec{0}$. So, *Assumption 4 holds here with $\alpha = 0$ for any FL algorithm.*

In fact, the above analysis and result (i.e., $\alpha = 0$) can be extended to networks whose training dynamics follow that of a linearized model, which has been shown to be the case for infinite-width networks (see for e.g., Lee et al. [2019] and Jacot et al. [2018]) and has been also used on

applications for finite-width networks (for e.g., in Mu et al. [2020]).

We now present the abridged version of the convergence result of `FedGLOMO`, followed by some important remarks. Its full version and detailed proof are in Appendix A and G.1, respectively.

**Theorem 1 (Smooth non-convex).** *Let Assumptions 1, 2 and 3 hold. Further, suppose Assumption 4 is true for* `FedGLOMO`*. In* `FedGLOMO`*, for each round* $k$*, set* $\eta_k = \eta = \mathcal{O}(\frac{1}{LEK^{1/3}C^{1/3}})$*, where* $C = \mathcal{O}\big(\max\big(\frac{\alpha}{n}, \frac{E^2(1+q)^2}{r}\big)\big)$*, and* $\beta_k = \mathcal{O}((1+q)\eta^2 L^2 E^4)$*. Suppose we use full-device participation (i.e., the global batch size is* $n$*) only at* $k = 0$*. Then,* `FedGLOMO` *can achieve* $\mathbb{E}_{k^* \sim \text{Unif}[0,K-1]}[\|\nabla f(\boldsymbol{w}_{k^*})\|^2] \leq \epsilon$ *in* $K = \mathcal{O}\big(\max\big(\sqrt{\frac{\alpha}{n}}, \frac{1+q}{\sqrt{r}}\big)\epsilon^{-1.5}\big)$ *rounds of communication and* $E = \mathcal{O}(1)$ *local steps.*

**Remark 1 (Better iteration complexity).** As per Theorem 1, for converging to an $\epsilon$-stationary point, `FedGLOMO` needs $T = KE$ to be $\mathcal{O}\big(\max\big(\sqrt{\frac{\alpha}{n}}, \frac{1}{\sqrt{r}}\big)\epsilon^{-1.5}\big)$. This iteration complexity is the same as that of `MimeMVR` (Karimireddy et al. [2020]) *but without using the bounded client dissimilarity assumption*, i.e. eq. (2), (also see the next remark for more details on this) and better than other related works in the federated setting; see Table 1. We underscore the significance of global momentum here by comparing this complexity of `FedGLOMO` to that of `FedLOMO` (recall this is a simpler version of `FedGLOMO` with only local momentum and *no* global momentum, described in Appendix F) under partial-device participation and compression which is $\mathcal{O}(\frac{1}{r}\epsilon^{-2})$; see Theorem 3 in the Appendix.

**Remark 2 (No requirement of bounded client dissimilarity (BCD) assumption).** Divergent from related works, Theorem 1 *does not use* the commonly used BCD assumption, i.e., eq. (2). This is achieved by utilizing the smoothness and non-negativity of the $f_i$'s, specifically $\frac{1}{n}\sum_{i\in[n]}\|\nabla f_i(\boldsymbol{w})\|^2 \leq \frac{1}{n}\sum_{i\in[n]}2L(f_i(\boldsymbol{w}) - f_i^*) \leq 2Lf(\boldsymbol{w})$; see the proof outline of Theorem 1 in Appendix A. Instead of the BCD assumption, we use our empirically verified Assumption 4 to provide a tighter (when $\alpha \ll n$) and data-dependent convergence result. Note that Assumption 4 will always hold for some $\alpha \leq n$, regardless of the degree of client heterogeneity. Thus, Theorem 1 allows for *arbitrary client heterogeneity*.

**Remark 3 (Compressed communication).** To our knowledge, `FedGLOMO` is the *first algorithm* that attains the aforementioned improved iteration complexity for FL on smooth non-convex functions *with compressed communication*. We emphasize that the choice of quantities compressed in line 10 of Algorithm 2 is important. This particular choice enables deriving the improved complexity by first deriving a result analogous to smoothness, i.e., $\|(\boldsymbol{w}_k - \boldsymbol{w}_{k,E}^{(i)}) - (\boldsymbol{w}_{k-1} - \widehat{\boldsymbol{w}}_{k-1,E}^{(i)})\| \leq \widehat{L}\|\boldsymbol{w}_k - \boldsymbol{w}_{k-1}\|$

(see Lemma 9 in Appendix G.1). The straightforward choice of sending $Q_D(\boldsymbol{w}_k - \boldsymbol{w}_{k,E}^{(i)})$ and $Q_D(\boldsymbol{w}_{k-1} - \widehat{\boldsymbol{w}}_{k-1,E}^{(i)})$ prohibits us from deriving the improved rate, unless we also assume $Q_D(.)$ to be a Lipschitz operator.

In Appendix B, for $r \ll n$, we show that using the quantization scheme of Alistarh et al. [2017] with $s = \sqrt{d}$, `FedGLOMO` achieves more than a five-fold saving in the *total* communication cost as compared to when there is full-precision communication in `FedGLOMO`.

**Remark 4 (A limitation).** Even though our iteration complexity of $T = \mathcal{O}(\epsilon^{-1.5})$ is better than that of `FedCOMGATE` proposed by Haddadpour et al. [2021] (which is $\mathcal{O}(\epsilon^{-2})$), our communication complexity of $K = \mathcal{O}(\epsilon^{-1.5})$ is higher than that theirs which is $K = \mathcal{O}(\epsilon^{-1})$ (albeit under an extra assumption on the quantizer, namely Assumption 5 in their paper). Ideally, we would like to have $E = \mathcal{O}(\epsilon^{-p})$ and $K = \mathcal{O}(\epsilon^{-(1.5-p)})$ for some $p > 0$, in order to reduce `FedGLOMO`'s communication complexity. Exploring whether such a result is obtainable with our proposed style of momentum is an interesting future direction.

# 6 EXPERIMENTS

To show the efficacy of *global* momentum in `FedGLOMO`, we compare it against `FedLOMO` (recall this has only local momentum and no global momentum; see Appendix F) and `FedAvg` (McMahan et al. [2017]) with the standard momentum available in PyTorch applied to (i) only its local updates, and (ii) both local and global updates – all with compressed client-to-server communication. We denote (i) and (ii) by `FedAvg`-lm and `FedAvg`-glm ("lm" and "glm" stand for local momentum, and global + local momentum), respectively. `FedAvg` *with compression* is referred to as `FedPAQ` (Reisizadeh et al. [2020]). Similarly, we call `FedAvg`-lm and `FedAvg`-glm *with compression*, as `FedPAQ`-lm and `FedPAQ`-glm. We also compare against `FedCOMGATE` (Haddadpour et al. [2021]) which uses gradient tracking to *theoretically* derive a better communication-complexity than us (see Remark 4). For compression, the "qsgd" operator proposed in Alistarh et al. [2017] is used.

We consider the task of classification on CIFAR-10 and Fashion-MNIST (Xiao et al. [2017]) abbreviated as FM-NIST henceforth. The model used is a two-layer neural network with ReLU activation in the hidden layers. The size of both the hidden layers is 300/600 for FMNIST/CIFAR-10. We train the models using the categorical cross-entropy loss with $\ell_2$-regularization. The weight decay value in PyTorch (to apply $\ell_2$-regularization) is set to 1e-4. We consider both homogeneous and heterogeneous data distribution among the clients. Similar to McMahan et al. [2017], for the heterogeneous case, we distribute the data among the clients such that each client can have data from either one or (at most) two classes – note that this is a high degree of heterogeneity. The exact procedure is described in Appendix E. The num-

ber of clients ($n$) in all the experiments is set to 50, with each client having the same number of samples. The global batch-size $r$ is 25, and the number of local updates per round (i.e., $E$) is 10. All full gradients are replaced by stochastic gradients computed on a (per-client) batch size of 256. The learning rates, momentum parameters of the algorithms, and some other experimental details are in Appendix E.

In Fig. 1, we compare `FedPAQ-lm`, `FedPAQ-glm`, `FedLOMO` and `FedCOMGATE` with 4 (resp., 8) bits per-round against `FedGLOMO` with 2 (resp., 4) bits per-round on FMNIST (resp., CIFAR-10) in the heterogeneous and homogeneous cases. We set the number of per-round bits used by `FedGLOMO` to be half the number used by all other algorithms, so that each one has the same *per-round* communication budget. All plots depict results over 3 independent runs; the shaded regions represent $\pm 1$ standard deviation whereas the solid lines are the respective means. Please see the discussion in the figure caption. These results illustrate the *power of global momentum*.

Next, in the *no-compression heterogeneous* case, we compare against `Mime` (specifically, "`MimeSGDm`") of Karimireddy et al. [2020] which also attains a complexity of $\mathcal{O}(\epsilon^{-1.5})$ but without compressed communication, and is tailored to handle client heterogeneity. Having shown the suboptimality of `FedLOMO` and `FedPAQ-lm` in Fig. 1, we only compare `FedAvg-glm`, `FedGLOMO` without compression and `MimeSGDm` in the heterogeneous case in Fig. 2. The plots in Fig. 2 show that the implicit client-drift controlling ability of our proposed global momentum is on par with the explicit client-drift controlling mechanism of `Mime`. The test error values averaged over the last five rounds for the plots in Figures 1 and 2 are in Tables 2 and 3, respectively.

We also provide some more empirical results on CIFAR-100 in Appendix E.1.

| Algo. | CIFAR-10 Het. | FMNIST Het. |
|---|---|---|
| `FedPAQ-lm` | $50.26 \pm 0.85$ | $16.17 \pm 0.53$ |
| `FedPAQ-glm` | $49.88 \pm 1.15$ | $15.87 \pm 1.10$ |
| `FedLOMO` | $53.74 \pm 0.17$ | $18.95 \pm 0.19$ |
| `FedGLOMO` | $\mathbf{46.42 \pm 0.05}$ | $\mathbf{13.55 \pm 0.32}$ |
| `FedCOMGATE` | $\mathbf{46.26 \pm 0.25}$ | $15.32 \pm 0.09$ |
| Algo. | CIFAR-10 Hom. | FMNIST Hom. |
| `FedPAQ-lm` | $\mathbf{45.13 \pm 0.07}$ | $13.08 \pm 0.05$ |
| `FedPAQ-glm` | $45.70 \pm 0.10$ | $11.76 \pm 0.06$ |
| `FedLOMO` | $45.96 \pm 0.01$ | $14.22 \pm 0.01$ |
| `FedGLOMO` | $\mathbf{44.97 \pm 0.05}$ | $\mathbf{10.98 \pm 0.05}$ |
| `FedCOMGATE` | $45.46 \pm 0.03$ | $12.24 \pm 0.01$ |

Table 2: Average **test error** % ($\pm$ standard deviation) over the last five rounds for the plots in the *heterogeneous* (*top*) and *homogeneous* (*bottom*) cases in Figure 1.

| Algo. | CIFAR-10 Het. | FMNIST Het. |
|---|---|---|
| `FedAvg-glm` | $50.26 \pm 0.74$ | $16.17 \pm 0.53$ |
| `MimeSGDm` | $46.10 \pm 0.13$ | $\mathbf{13.34 \pm 0.25}$ |
| `FedGLOMO` | $\mathbf{45.41 \pm 0.15}$ | $\mathbf{13.48 \pm 0.26}$ |

Table 3: Average **test error** % ($\pm$ standard deviation) over the last five rounds for the plots in Figure 2.

**Verifying Assumption 4 for `FedGLOMO`:** For each round $k$, we compute $\alpha = \max_{\tau \in [E]} \frac{\| \sum_{i \in [n]} \widetilde{e}_{k,\tau}^{(i)} \|^2}{\sum_{i \in [n]} \| \widetilde{e}_{k,\tau}^{(i)} \|^2}$, where $\widetilde{e}_{k,\tau}^{(i)}$ is as defined in Assumption 4, for 4 and 2 bit `FedGLOMO` on CIFAR-10 and FMNIST, respectively. Note that we remove the expectation (w.r.t. the stochastic gradients) while computing $\alpha$ for empirical verification. In Fig. 3, we plot $(\alpha/n)$ over different rounds for the heterogeneous as well as homogeneous case on both datasets; see the discussion in the figure caption.

# 7 CONCLUSION

We presented `FedGLOMO`, a communication-efficient algorithm for faster federated learning via the application of variance-reducing momentum, both in the aggregation step at the server as well as local client updates. We showed that `FedGLOMO` has better iteration complexity than prior work on smooth non-convex functions with compressed communication. Further, unlike prior work, our result does not use the bounded client dissimilarity assumption, even holding under arbitrary client heterogeneity. We also demonstrate the efficacy of `FedGLOMO` via extensive experiments.

**Acknowledgements**

This work is supported by NSF grants CCF-1564000, IIS-1546452 and HDR-1934932, AFOSR grant FA9550-19-1-0005, and NASA grant 80NSSC21M0071.

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

Yossi Arjevani, Yair Carmon, John C Duchi, Dylan J Foster, Nathan Srebro, and Blake Woodworth. Lower bounds

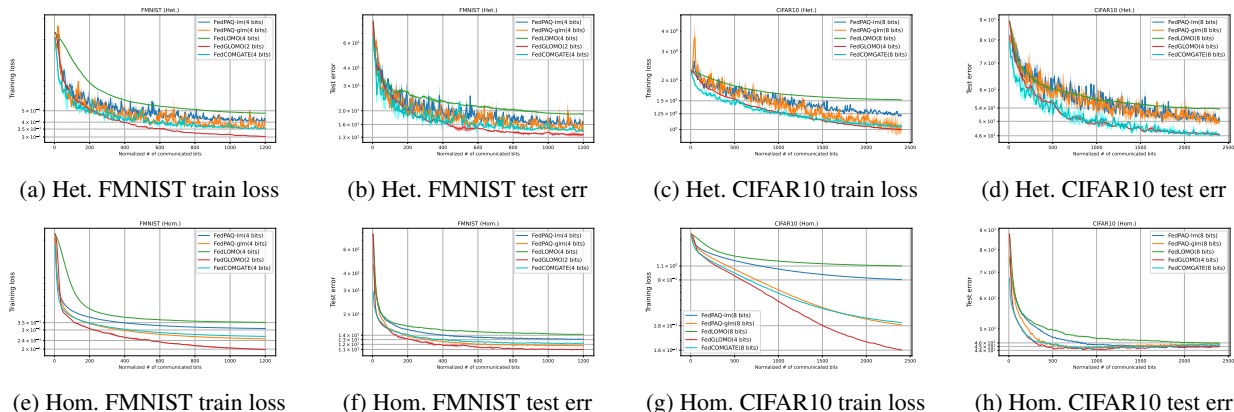

(a) Het. FMNIST train loss    (b) Het. FMNIST test err    (c) Het. CIFAR10 train loss    (d) Het. CIFAR10 test err

(e) Hom. FMNIST train loss    (f) Hom. FMNIST test err    (g) Hom. CIFAR10 train loss    (h) Hom. CIFAR10 test err

Figure 1: Comparison of `FedPAQ-lm`, `FedPAQ-glm`, `FedLOMO`, `FedGLOMO` and `FedCOMGATE` (Haddadpour et al. [2021]) with the same per-round communication budget on FMNIST and CIFAR-10 in the heterogeneous (top four figs.) and homogeneous (bottom four figs.) settings, respectively. The x-axis is the total number of communicated bits divided by the dimension $d$ and the global batch-size $r$. `FedGLOMO` is the **fastest** and most **communication-efficient** algorithm in almost all the cases; for e.g., in the heterogeneous case for both datasets, `FedGLOMO` attains the final test error of `FedPAQ-glm` (resp., `FedPAQ-lm`) with less than a **half** (resp., only about a **third**) of the number of bits used by `FedPAQ-glm` (resp., `FedPAQ-lm`). Further, `FedGLOMO` and `FedLOMO` have a smoother trajectory than other algorithms in the heterogeneous case due to variance-reducing momentum. Observe that `FedLOMO` and `FedPAQ-lm` (with only local momentum) are slower than `FedGLOMO` and `FedPAQ-slm` (with both local and global momentum), showing the ineffectiveness of only local momentum and **the power of combining both local and global momentum**. Also, note that `FedGLOMO` performs much better than `FedCOMGATE` in the homogeneous case.

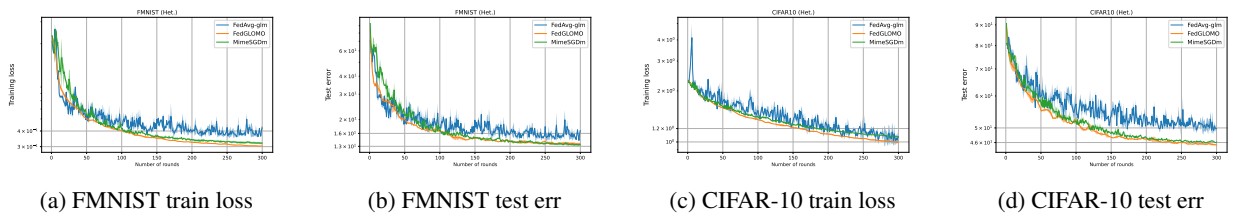

(a) FMNIST train loss    (b) FMNIST test err    (c) CIFAR-10 train loss    (d) CIFAR-10 test err

Figure 2: Comparison of `FedAvg-glm`, `FedGLOMO` (without compression) and `MimeSGDm` on FMNIST and CIFAR-10 in the **heterogeneous** case. On both datasets, `FedAvg-glm` is the slowest while `FedGLOMO` is somewhat faster than `MimeSGDm`. While `Mime` has an explicit client-drift control mechanism, we do not have that in `FedGLOMO`, but still **our proposed global momentum implicitly mitigates client-drift** as well as `Mime`.

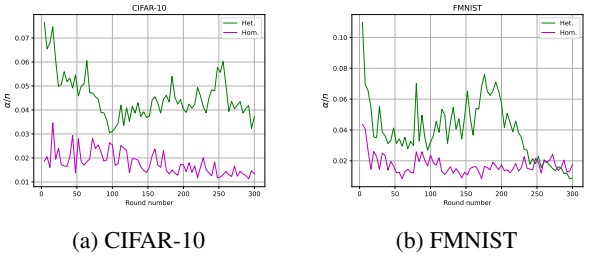

(a) CIFAR-10    (b) FMNIST

Figure 3: Variation of $\left(\frac{\alpha}{n}\right)$ over different rounds of 4 and 2 bit `FedGLOMO` for CIFAR-10 (Fig. 3a) and FMNIST (Fig. 3b) in the heterogeneous and homogeneous cases. In both cases, notice that $\alpha \ll n$ throughout training. Also, as discussed after the statement of Assumption 4, note that $\left(\frac{\alpha}{n}\right)$ is higher for the heterogeneous case (except at the end of training for FMNIST). See Figure 4 in the Appendix for the same on `FedAvg`.

for non-convex stochastic optimization. *arXiv preprint arXiv:1912.02365*, 2019.

Debraj Basu, Deepesh Data, Can Karakus, and Suhas Diggavi. Qsparse-local-sgd: Distributed sgd with quantiza-

tion, sparsification and local computations. In *Advances in Neural Information Processing Systems*, pages 14695–14706, 2019.

Ahmed Khaled Ragab Bayoumi, Konstantin Mishchenko, and Peter Richtárik. Tighter theory for local sgd on identical and heterogeneous data. In *International Conference on Artificial Intelligence and Statistics*, pages 4519–4529, 2020.

Jeremy Bernstein, Yu-Xiang Wang, Kamyar Azizzadenesheli, and Anima Anandkumar. signsgd: Compressed optimisation for non-convex problems. *arXiv preprint arXiv:1802.04434*, 2018.

Yiuye Chen, Abolfazl Hashemi, and Haris Vikalo. Communication-efficient algorithms for decentralized optimization over directed graphs. *arXiv preprint arXiv:2005.13189*, 2020.

Yiyue Chen, Abolfazl Hashemi, and Haris Vikalo. Communication-efficient variance-reduced decentralized stochastic optimization over time-varying directed graphs. *IEEE Transactions on Automatic Control*, 2021.

Ashok Cutkosky and Francesco Orabona. Momentum-based variance reduction in non-convex sgd. In *Advances in Neural Information Processing Systems*, pages 15236–15245, 2019.

Aaron Defazio, Francis Bach, and Simon Lacoste-Julien. Saga: A fast incremental gradient method with support for non-strongly convex composite objectives. *Advances in neural information processing systems*, 27, 2014.

Cong Fang, Chris Junchi Li, Zhouchen Lin, and Tong Zhang. Spider: Near-optimal non-convex optimization via stochastic path-integrated differential estimator. In *Advances in Neural Information Processing Systems*, pages 689–699, 2018.

Eduard Gorbunov, Konstantin Burlachenko, Zhize Li, and Peter Richtárik. Marina: Faster non-convex distributed learning with compression. *arXiv preprint arXiv:2102.07845*, 2021.

Farzin Haddadpour, Mohammad Mahdi Kamani, Aryan Mokhtari, and Mehrdad Mahdavi. Federated learning with compression: Unified analysis and sharp guarantees. *arXiv preprint arXiv:2007.01154*, 2021.

Abolfazl Hashemi, Anish Acharya, Rudrajit Das, Haris Vikalo, Sujay Sanghavi, and Inderjit S Dhillon. On the benefits of multiple gossip steps in communication-constrained decentralized federated learning. *IEEE Transactions on Parallel and Distributed Systems*, 2021.

Samuel Horváth, Dmitry Kovalev, Konstantin Mishchenko, Sebastian Stich, and Peter Richtárik. Stochastic distributed learning with gradient quantization and variance reduction. *arXiv preprint arXiv:1904.05115*, 2019.

Zhouyuan Huo, Qian Yang, Bin Gu, Lawrence Carin Huang, et al. Faster on-device training using new federated momentum algorithm. *arXiv preprint arXiv:2002.02090*, 2020.

Arthur Jacot, Franck Gabriel, and Clément Hongler. Neural tangent kernel: Convergence and generalization in neural networks. *Advances in neural information processing systems*, 31, 2018.

Rie Johnson and Tong Zhang. Accelerating stochastic gradient descent using predictive variance reduction. *Advances in neural information processing systems*, 26, 2013.

Sai Praneeth Karimireddy, Satyen Kale, Mehryar Mohri, Sashank J Reddi, Sebastian U Stich, and Ananda Theertha Suresh. Scaffold: Stochastic controlled averaging for federated learning. *arXiv preprint arXiv:1910.06378*, 2019.

Sai Praneeth Karimireddy, Martin Jaggi, Satyen Kale, Mehryar Mohri, Sashank J Reddi, Sebastian U Stich, and Ananda Theertha Suresh. Mime: Mimicking centralized stochastic algorithms in federated learning. *arXiv preprint arXiv:2008.03606*, 2020.

Prashant Khanduri, Pranay Sharma, Haibo Yang, Mingyi Hong, Jia Liu, Ketan Rajawat, and Pramod Varshney. Stem: A stochastic two-sided momentum algorithm achieving near-optimal sample and communication complexities for federated learning. *Advances in Neural Information Processing Systems*, 34, 2021.

Anastasia Koloskova, Nicolas Loizou, Sadra Boreiri, Martin Jaggi, and Sebastian Stich. A unified theory of decentralized sgd with changing topology and local updates. In *International Conference on Machine Learning*, pages 5381–5393. PMLR, 2020.

Jaehoon Lee, Lechao Xiao, Samuel Schoenholz, Yasaman Bahri, Roman Novak, Jascha Sohl-Dickstein, and Jeffrey Pennington. Wide neural networks of any depth evolve as linear models under gradient descent. *Advances in neural information processing systems*, 32, 2019.

Tian Li, Anit Kumar Sahu, Manzil Zaheer, Maziar Sanjabi, Ameet Talwalkar, and Virginia Smith. Federated optimization in heterogeneous networks. *arXiv preprint arXiv:1812.06127*, 2018.

Xiang Li, Kaixuan Huang, Wenhao Yang, Shusen Wang, and Zhihua Zhang. On the convergence of fedavg on non-iid data. *arXiv preprint arXiv:1907.02189*, 2019.

Xianfeng Liang, Shuheng Shen, Jingchang Liu, Zhen Pan, Enhong Chen, and Yifei Cheng. Variance reduced local sgd with lower communication complexity. *arXiv preprint arXiv:1912.12844*, 2019.

Yujun Lin, Song Han, Huizi Mao, Yu Wang, and William J Dally. Deep gradient compression: Reducing the communication bandwidth for distributed training. *arXiv preprint arXiv:1712.01887*, 2017.

Deyi Liu, Lam M Nguyen, and Quoc Tran-Dinh. An optimal hybrid variance-reduced algorithm for stochastic composite nonconvex optimization. *arXiv preprint arXiv:2008.09055*, 2020.

Brendan McMahan, Eider Moore, Daniel Ramage, Seth Hampson, and Blaise Aguera y Arcas. Communication-efficient learning of deep networks from decentralized data. In *Artificial Intelligence and Statistics*, pages 1273–1282. PMLR, 2017.

Fangzhou Mu, Yingyu Liang, and Yin Li. Gradients as features for deep representation learning. *arXiv preprint arXiv:2004.05529*, 2020.

Lam M Nguyen, Jie Liu, Katya Scheinberg, and Martin Takáč. Sarah: A novel method for machine learning problems using stochastic recursive gradient. In *International Conference on Machine Learning*, pages 2613–2621. PMLR, 2017.

Kumar Kshitij Patel and Aymeric Dieuleveut. Communication trade-offs for synchronized distributed sgd with large step size. *arXiv preprint arXiv:1904.11325*, 2019.

Shi Pu and Angelia Nedić. Distributed stochastic gradient tracking methods. *Mathematical Programming*, pages 1–49, 2020.

Zhaonan Qu, Kaixiang Lin, Jayant Kalagnanam, Zhaojian Li, Jiayu Zhou, and Zhengyuan Zhou. Federated learning's blessing: Fedavg has linear speedup. *arXiv preprint arXiv:2007.05690*, 2020.

Sashank Reddi, Zachary Charles, Manzil Zaheer, Zachary Garrett, Keith Rush, Jakub Konečnỳ, Sanjiv Kumar, and H Brendan McMahan. Adaptive federated optimization. *arXiv preprint arXiv:2003.00295*, 2020.

Amirhossein Reisizadeh, Aryan Mokhtari, Hamed Hassani, Ali Jadbabaie, and Ramtin Pedarsani. Fedpaq: A communication-efficient federated learning method with periodic averaging and quantization. In *International Conference on Artificial Intelligence and Statistics*, pages 2021–2031, 2020.

Sebastian U Stich. Local sgd converges fast and communicates little. *arXiv preprint arXiv:1805.09767*, 2018.

Sebastian U Stich and Sai Praneeth Karimireddy. The error-feedback framework: Better rates for sgd with delayed gradients and compressed communication. *arXiv preprint arXiv:1909.05350*, 2019.

Sebastian U Stich, Jean-Baptiste Cordonnier, and Martin Jaggi. Sparsified sgd with memory. In *Advances in Neural Information Processing Systems*, pages 4447–4458, 2018.

Ananda Theertha Suresh, X Yu Felix, Sanjiv Kumar, and H Brendan McMahan. Distributed mean estimation with limited communication. In *International Conference on Machine Learning*, pages 3329–3337, 2017.

Hanlin Tang, Shaoduo Gan, Ce Zhang, Tong Zhang, and Ji Liu. Communication compression for decentralized training. In *Advances in Neural Information Processing Systems*, pages 7652–7662, 2018.

Jianyu Wang and Gauri Joshi. Cooperative sgd: A unified framework for the design and analysis of communication-efficient sgd algorithms. *arXiv preprint arXiv:1808.07576*, 2018.

Jianyu Wang, Vinayak Tantia, Nicolas Ballas, and Michael Rabbat. Slowmo: Improving communication-efficient distributed sgd with slow momentum. *arXiv preprint arXiv:1910.00643*, 2019.

Blake Woodworth, Kumar Kshitij Patel, Sebastian U Stich, Zhen Dai, Brian Bullins, H Brendan McMahan, Ohad Shamir, and Nathan Srebro. Is local sgd better than mini-batch sgd? *arXiv preprint arXiv:2002.07839*, 2020.

Jiaxiang Wu, Weidong Huang, Junzhou Huang, and Tong Zhang. Error compensated quantized sgd and its applications to large-scale distributed optimization. *arXiv preprint arXiv:1806.08054*, 2018.

Han Xiao, Kashif Rasul, and Roland Vollgraf. Fashion-mnist: a novel image dataset for benchmarking machine learning algorithms. *arXiv preprint arXiv:1708.07747*, 2017.

Hao Yu, Sen Yang, and Shenghuo Zhu. Parallel restarted sgd for non-convex optimization with faster convergence and less communication. *arXiv preprint arXiv:1807.06629*, 2 (4):7, 2018.

Dongruo Zhou, Pan Xu, and Quanquan Gu. Stochastic nested variance reduced gradient descent for nonconvex optimization. *Advances in neural information processing systems*, 2018.

Martin Zinkevich, Markus Weimer, Lihong Li, and Alex Smola. Parallelized stochastic gradient descent. *Advances in neural information processing systems*, 23:2595–2603, 2010.
