# OpenReview forum: "Faster Non-Convex Federated Learning via Global and Local Momentum"
_auai.org/UAI/2022/Conference — UAI 2022 Poster_

### Official Review · Reviewer_r9CH · 2022-04-13

**Q2(1) Originality/Novelty:** 3
**Q2(2) Significance/Impact:** 3
**Q2(3) Correctness/Technical Quality:** 3
**Q2(6) Clarity Of Writing:** 3
**Q6 Overall Score:** 5
**Q8 Confidence In Your Score:** 2

**Q1 Summary And Contributions:**

The paper under consideration is devoted to the federated learning algorithm. The authors propose a new algorithm called FedGLOMO where additional global momentum is introduced. The proposed version achieves better than already known complexity.

**Q2 Assessment Of The Paper:**

More detailed information regarding each of these aspects is given below:

**Q2(4) Quality Of Experiments (Optional):**

3: Good: The experimental evaluation is adequate, and the results convincingly support the main claims.

**Q2(5) Reproducibility:**

3: Good: Key resources (e.g., proofs, code, data) are available and key details (e.g., proofs, experimental setup) are sufficiently well-described for competent researchers to confidently reproduce the main results.

**Q3 Main Strengths:**

- A new algorithm with better complexity is proposed.
- The algorithm can be applied to many ML problems.

**Q4 Main Weakness:**

The theoretical results are obtained by introducing new Assumption 4. I'm not very familiar with federated learning and it is hard to judge for me how restrictive it is.

**Q5 Detailed Comments To The Authors:**

I find the paper well written. There are places where the statement could be improved. For example in Definition 3.1 twice differentiability is not needed. I recommend to check the paper once again.

**Q7 Justification For Your Score:**

I'm not sure about the significance of the paper. For me, it is not clear how restrictive are new assumptions.

**Q9 Complying With Reviewing Instructions:**

1: Yes.

---

### Official Review · Reviewer_FHbV · 2022-04-14

**Q2(1) Originality/Novelty:** 3
**Q2(2) Significance/Impact:** 3
**Q2(3) Correctness/Technical Quality:** 3
**Q2(6) Clarity Of Writing:** 3
**Q6 Overall Score:** 6
**Q8 Confidence In Your Score:** 4

**Q1 Summary And Contributions:**

This paper studies nonconvex and smooth optimization in federated learning setting. The authors propose an algorithm with both local variance reduced momentum and global variance reduced momentum. They also proposes a new assumption on the gradient dependence of each worker machines. They prove that the proposed algorithm achieve the best iteration complexity and compare it in experiments with baselines.


**Q2 Assessment Of The Paper:**

More detailed information regarding each of these aspects is given below:

**Q2(4) Quality Of Experiments (Optional):**

3: Good: The experimental evaluation is adequate, and the results convincingly support the main claims.

**Q2(5) Reproducibility:**

3: Good: Key resources (e.g., proofs, code, data) are available and key details (e.g., proofs, experimental setup) are sufficiently well-described for competent researchers to confidently reproduce the main results.

**Q3 Main Strengths:**

This paper is well written. The algorithms and theoretical results are presented in an accessible way and discussed in detail. The results seem to be competitive.

**Q4 Main Weakness:**

It should be discussed in more detail how this result matches that of existing work in different settings with different problem-dependent parameters.

**Q5 Detailed Comments To The Authors:**

This paper studies nonconvex and smooth optimization in a federated learning setting. The authors propose an algorithm with both local variances reduced momentum and global variance reduced momentum. They also propose a new assumption on the gradient dependence of each worker machine. They prove that the proposed algorithm achieves the best iteration complexity and compare it in experiments with baselines.

This paper is well written. The algorithms and theoretical results are presented in an accessible way and discussed in detail. The results seem to be competitive.
It should be discussed the relationship between r and n/\alpha when presenting the complexity. This should also be noted in Table 1 that the complexity does not exactly match that of Karimireddy et al. [2020].

I am not sure when Assumption 4 actually holds in practice with a small \alpha = o(n). Note that epsilon depends on 1/n \sum_{j} w^{(j)} for all agents. This means all \epsilon could be dependent in some way and thus \alpha = Cn for some small constant but should be the same order as n.


**Q7 Justification For Your Score:**

Overall, the contribution of this paper is solid and the paper is well written.

**Q9 Complying With Reviewing Instructions:**

1: Yes.

---

### Official Review · Reviewer_M24P · 2022-04-14

**Q2(1) Originality/Novelty:** 2
**Q2(2) Significance/Impact:** 3
**Q2(3) Correctness/Technical Quality:** 3
**Q2(6) Clarity Of Writing:** 2
**Q6 Overall Score:** 6
**Q8 Confidence In Your Score:** 4

**Q1 Summary And Contributions:**

This paper proposes FedGLOMO, a new federated learning algorithm with a better complexity than previous work for smooth non-convex functions. FedGLOMO uses the variance reduction approach with global and local momentum.

**Q2 Assessment Of The Paper:**

More detailed information regarding each of these aspects is given below:

**Q2(4) Quality Of Experiments (Optional):**

3: Good: The experimental evaluation is adequate, and the results convincingly support the main claims.

**Q2(5) Reproducibility:**

3: Good: Key resources (e.g., proofs, code, data) are available and key details (e.g., proofs, experimental setup) are sufficiently well-described for competent researchers to confidently reproduce the main results.

**Q3 Main Strengths:**

FedGLOMO is communication-efficient that integrates variance-reduction, both at the server and at local client updates. It also has a good iteration complexity than prior works on non-convex functions with compressed communication. Experiments show the efficacy of the proposed method.

**Q4 Main Weakness:**

Assumption 4 is not standard. (See the comment below).

**Q5 Detailed Comments To The Authors:**

Although Assumption 4 is not standard and the authors claimed that it holds for alpha = n, I do not see why alpha is necessary in the analysis. As far as I understand, the analysis still hold at the same magnitude of epsilon when alpha = n. Is this true? I think it is beneficial if the authors add a discussion on how the theoretical analysis changes when alpha = n.

Another minor thing: The fonts of algorithm names are not consistent. (E.g. Title of section 4 had some error in changing the fonts)

**Q7 Justification For Your Score:**

Though I still have concern, I think the contributions are good.

---
The authors replied and resolved my concerns. Hence I updated my decision to 6: Weak Accept: Technically solid, moderate to high impact paper.

**Q9 Complying With Reviewing Instructions:**

1: Yes.

---

### Decision · Program_Chairs · 2022-05-15

**Decision:**

Accept (Poster)

**Comment:**

Meta Review: The authors have addressed the reviewers’ concerns. All the reviewers are not against the publication of the paper. Please add the discussion with the reviewer into the final version, especially the discussion on Assumption 4. The authors should also add the new empirical results too.